# Higher operating theatre temperature during burn surgery increases physiological heat strain, subjective workload, and fatigue of surgical staff

**Zehra Palejwala**[1]*, **Karen E. Wallman**[1°], **Shane Maloney**[1°], **Grant J. Landers**[1°], **Ullrich K. H. Ecker**[2], **Mark W. Fear**[3], **Fiona M. Wood**[4]

1 School of Human Sciences, The University of Western Australia, Crawley, Western Australia, Australia,
2 School of Psychological Science, The University of Western Australia, Crawley, Western Australia,
Australia, 3 School of Biomedical Sciences, University of Western Australia, Crawley, Western Australia,
Australia, 4 Burn Injury Research Unit, Burn Service of WA South Metropolitan Health Service, University of
Western Australia, Crawley, Western Australia, Australia

☯ These authors contributed equally to this work.
* Zehra.palejwala@research.uwa.edu.au

**Data Availability Statement:** All relevant data are within the paper and its Supporting information files.

## Abstract

Raising the ambient temperature of the operating theatre is common practice during burn surgeries to maintain the patient's core body temperature; however, the effects of operating in the heat on cognitive performance, manual dexterity, and perceived workload of surgical staff have not been assessed in a real-world context. Therefore, the aim was to assess the real-time impact of heat during burn surgeries on staff's cognitive function, manual dexterity, and perceptual measures (workload, thermal sensation, thermal comfort, perceived exertion, and fatigue) and physiological parameters (core temperature, heart-rate, fluid loss, and dehydration). Ten burn surgery staff members were assessed in CON (24.0±1.1°C, 45±6% relative humidity [RH]) and HOT (30.8±1.6°C, 39±7% RH) burn surgeries (average 150 min duration). Cognitive performance, manual dexterity, and perceptual measures were recorded pre- and post-surgery, while physiological parameters were recorded throughout surgery. HOT conditions did not significantly affect manual dexterity or cognitive function ($p > .05$), however HOT resulted in heat strain (increased heart-rate, core temperature, and fluid loss: $p < .05$), and increased subjective workload, discomfort, perceived exertion, and fatigue compared to CON conditions ($p < .05$). Cognitive function and manual dexterity were maintained in hot conditions, suggesting that operating in approximately 31°C heat is a safe approach for patient treatment. However, job burnout, which is positively correlated with perceived workload, and the impact of cumulative fatigue on the mental health of surgery staff, must be considered in the context of supporting an effective health workforce.

**Funding:** Funding - This study was funded by the Fiona Wood Foundation. Funding was received by ZP. There was no grant number. URL: https://www.fionawoodfoundation.com/ The funders had no role in study design, data collection and analyses, decision to publish, or preparation of the manuscript.

**Competing interests:** The authors have declared that no competing interests exist.

## Introduction

Major burn surgeries (usually $\geq$ 20% total body surface area [TBSA]) are typically conducted in ambient temperatures of 30–40°C to prevent patients from developing intraoperative hypothermia [1]. This can improve patient outcomes; however, patient outcomes also depend on the cognitive function [2], manual dexterity/technical skills [3], and fatigue levels of surgical teams [4].

Heat exposure can lead to heat and cardiovascular strain [5], and dehydration if fluids are not adequately replaced, all of which can impair physical and cognitive function [6–9], specifically complex decision-making tasks such as those involved during surgery [10, 11]. Dehydration, heat strain, and cardiovascular strain are further exacerbated when individuals wear personal protective equipment (PPE) in hot ambient conditions [12], which is a major concern for surgery staff.

Perceptions of workload are consistently higher in hot temperatures [13]. In both warm (26°C) and hot (34°C) compared to thermoneutral (19–23°C) operating theatres (OT), the physical, mental, and temporal demand of surgery tasks can increase [2, 14, 15], as well as the surgery team's subjective discomfort [16]. An increase in perceived workload is correlated with burnout, especially in the health care sector [17]. This is important as burnt-out employees often have poor mental health and an increased risk for cardiovascular disease [18, 19]. In a 2.5 h burn surgery simulation, executive functioning and verbal reasoning were impaired in a hot (34°C) compared to a cooler (23°C) OT [2], while manual dexterity scores tended to be lower [2, 16]. Together, these physical and cognitive effects represent a concern for surgery staff who have their own and the patients' welfare at risk. However, the impact of the heat on burn surgical teams has not been previously measured in a real-world (not-simulated) context.

Therefore, the aim of this study was to compare the impact of operating in hot ambient conditions (HOT) compared to control conditions (CON) on burn surgery staff, during real-life surgeries. It was hypothesised that operating in a hot theatre would result in heat strain, subsequently impairing manual dexterity and cognitive function, while increasing subjective workload and fatigue.

## Materials and methods

### Participants

Surgical staff from a burns department were recruited in the winter (June—October 2021; when average maximum ambient temperature was 20°C) to minimise the possibility of acclimatisation/acclimation, for testing in CON and HOT conditions. Descriptive statistics are provided in S1 Table. All staff gave written consent for participation in this study and patients gave either written or verbal consent (written consent was not able to be provided by all patients because of the nature and location of the burn injury). Verbal consent was witnessed by a member of the surgical burns team who then signed and dated the consent form, attesting that the requirements for informed consent were satisfied. Ethical approval was granted by the Human Research Ethics Committee of the University of Western Australia (2020/ET000239) and the South Metropolitan Health Service Human Research Ethics Committee (PRN RGS0000004250).

### Experimental design

The staff were assessed in thermoneutral (CON; 24°C) and heated (HOT; 31°C) surgeries, which all commenced between 8:30–9:30am. Ten staff members were recruited for participation in this study, of which seven were tested in both conditions. The remaining three were

tested in HOT only. There were 22 observations in the CON condition and 18 in HOT. Testing in the CON condition occasionally included two/three back-to-back cases, with staff remaining in theatre until the conclusion of the final case, to match testing times. No patients became hypothermic during surgery in either the CON or HOT condition. Staff wore the same standard surgical clothing and PPE (scrub gown, gloves, scrub hat, surgical mask) for each trial. S2 Table summarises the testing regime.

### Familiarisation session

Staff attended a familiarisation session approximately one week prior to being assessed during surgery, where they were made familiar with the questionnaires and practiced the performance tests to prevent a practice effect from occurring while testing [20]. All staff provided information on weekly heat exposure and physical activity via the International Physical Activity Questionnaire [21], to determine acclimatisation/acclimation status. The average, maximum, outdoor temperature during the testing period was 20˚C and no staff had travelled to a warmer climate in recent months prior to testing. Within the four months of testing, staff were exposed to a heated OT on 5 occurrences, with the average duration of exposure being 158 min, equating to a total average of 50 min per week. The average amount of recreational physical activity (not including job-related, indoor physical activity i.e. walking within the hospital) was 5 hours weekly, with most reporting light, as opposed to moderate and high intensity physical activity. Thus, no staff were determined to be acclimatised/acclimated. Female staff provided information on their menstrual cycle and contraception use so to determine differences in menstrual cycle phase during surgery. Height (cm) and body-mass (kg) were recorded and then the staff were familiarised with the testing equipment including heart-rate (HR) monitors (Polar RS400, Finland), digital platform weighing scales (SOEHNLE, Style sense comfort 100, Digital & Anko Glass Electronic), and the refractometer (for determining urine specific gravity; $U_{SG}$: ATAGO MASTER-URC/Na, Tokyo, Japan). Values obtained for $U_{SG}$ were classified as 'well hydrated' $<1.010$, 'minimal dehydration' = 1.010–1.020, 'significant dehydration' = 1.021–1.030 and 'serious dehydration' $>1.030$ [22]. It is important to note that measurement of USG may not reflect plasma osmolality [23], the most efficient measure to assess hydration status, and so the classifications provided may not be accurate in illustrating the extent of hypohydration. Staff were also provided with an ingestible core-temperature ($T_{CORE}$) pill (CorTemp, HQ Inc., Palmetto, USA).

### Protocol

Four to eight hours prior to surgery, staff ingested a pill that objectively measured $T_{CORE}$ [24]. Upon arrival to the hospital, staff provided a urine sample to determine $U_{SG}$. In private, nude body mass was measured to the nearest 0.1 kg using a digital platform scale (details provided above) and following this, staff were fitted with a HR monitor. In the OT, prior to surgery, both performance tests and all questionnaires, except the SURG-TLX, were completed. Staff then exited the OT (~2–3 min) to scrub before beginning surgery, and then fulfilled their usual roles within the OT. An initial (baseline) $T_{CORE}$ and HR measurements were taken as soon as surgery commenced and at 15-min intervals throughout surgery. Once surgery began, staff remained in the OT and did not consume any food/fluids until after the final measures were recorded at the conclusion of the surgery. All questionnaires and performance tasks were re-completed upon completion of surgery, followed by the assessment of nude body to determine fluid loss (pre nude body mass–post nude body mass) and the collection of a final urine sample.

## Perceptual questionnaires

Thermal sensation (TS) and thermal comfort (TC) were rated using 20-point scales from 'very cold' to 'very hot', and 'very comfortable' to 'very uncomfortable' [25], respectively. A score of 10 indicates optimal thermal sensation and comfort. Perceived exertion (RPE) was rated using the Borg 6-20-point scale which ranges from 'no exertion at all' to 'maximal exertion' [26]. Perceived workload was assessed using the SURG-TLX [27], a surgery-specific workload measure adapted from the NASA-TLX workload scale [28], which assesses workload over seven domains (mental demand, physical demand, temporal demand, task complexity, situational stress, distractions, and frustration) on a scale of 0 = very low to 100 = very high. With these instruments, a higher score indicates poorer health.

## Performance tests

The function of working memory–a core component of executive capacity–was measured by the counting span task (millisecond software) on a laptop [29]. In this test, participants are presented with cards featuring a number of both target dots (green: 3–9) and distractor dots (yellow: 3–9); participants count the number of target dots, press the corresponding key on a keyboard, and remember the count number. After a certain number of cards (starting with a set size of 3 and going up to 7, with two trials per set size), participants recall the counts (i.e., the number of dots they counted for each card) in order, starting with the first card (i.e., serial recall). The test ends when an individual fails to successfully recall the sequence on both trials of a particular set size (i.e., the number of cards presented depends on performance). Measures obtained from this task were the number of correct counting responses and counting latencies (in milliseconds [ms]), number of correct recall responses and recall latencies, and the counting span score (i.e., the highest span level at which participant correctly recalled 2 out of 3 sets). As such, measures included those that relate to basic counting performance as well as those assessing working memory capacity; for this reason, the term 'cognitive function' has been used to refer to these measures collectively. The highest number of correct responses that can be achieved for both the counting and recall tasks is 54; the highest possible counting span score is 7.

Manual dexterity was assessed by the Purdue pegboard task (60 s). In this test, participants were required to pick up one pin at a time and place as many pins in the holes of a board in 30 s, starting from the top hole and the dominant hand, progressing to the non-dominant hand once the 30 s were complete [30].

## Statistical analysis

Analysis was conducted using R Studio (Version 1.4.1717 for Windows). Linear mixed-models (within and between subjects) were used to assess all dependent variables, across all time points and in both conditions. All outputs were produced by running linear regression models (obtained using the *lmer* function) with random intercepts for individual participants, through the *anova* test function. This function removes missing observations, i.e. a complete case analysis was performed. One-way ANOVAs were used to assess differences in surgery duration and TBSA. Follow-up post hoc comparisons using *Tukey* adjustments were used. Significance was accepted at $p \leq .05$. All results presented within the written text and tables are expressed as mean ± SD and all figures are presented as individual data points or mean ± SEM. Cohen's *d* effect sizes (ES) with ±95% confidence intervals (CI) were also calculated, with effects $\geq 0.8$ representing large, 0.5–0.79 moderate, and $\leq 0.49$ small effects, respectively [31]. Only moderate to large ES are reported.

## Results

Environmental conditions were 24.0±1.1˚C, 45±6% RH for the CON trials, and 30.8±1.6˚C, 39 ±7% RH for the HOT trials. Surgery duration was not different between conditions (CON: 141 ± 50 min, HOT: 158 ± 51 min; $p$ = .287). Burn injury TBSA of patients was not different between conditions (CON: 8±13%, HOT: 20±7%; $p$ = .053). Of the seven females tested, three were post-menopausal, two were using an intrauterine device which meant that their menstrual cycle phase was unidentifiable, one was only tested once in the follicular phase, and one was in the follicular phase during the first testing session and the luteal phase during the second.

### Counting task

There was no effect of theatre temperature on counting latency $(p$ = .836); however, the main effect of time on counting latency approached significance $(p$ = .060), meaning that post-surgery, response times tended to be faster than pre-surgery. There was no effect of theatre temperature on number of correct responses $(p$ = .483), and there was no pre-/post-surgery difference on number of correct responses $(p$ = .427). There was no interaction between theatre temperature and time on counting latency $(p$ = .203; Fig 1) or the number of correct responses $(p$ = .757; Table 1).

### Recall task

There was no effect of theatre temperature on recall latency $(p$ = .623); however, there was a difference between recall latency pre and post-surgery $(p$ = .045), indicating that response times were faster post-surgery. There was no effect of theatre temperature on number of correct responses $(p$ = .964), and there was no pre-/post-surgery difference on number of correct responses $(p$ = .657). There was no effect of theatre temperature on overall counting span score $(p$ = .998), and no difference in scores pre and post-surgery $(p$ = .990). There was no

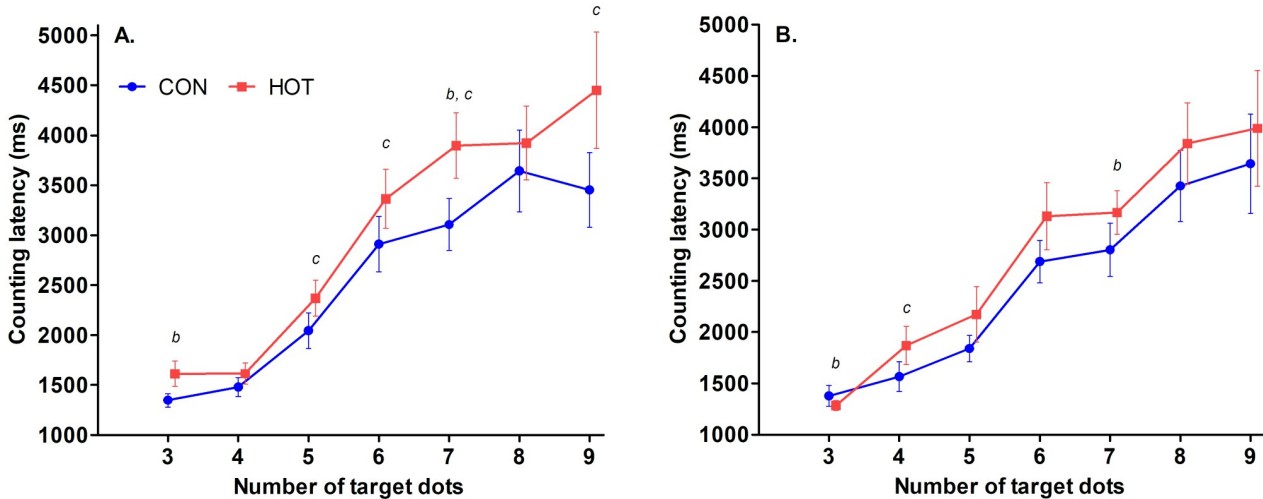

**Fig 1. Counting latencies 'pre' (A) and 'post' (B) surgery for target-dot numbers 3 to 9 in CON (Pre; n = 13, Post; n = 18) and HOT (n = 18) surgeries.** [b]indicates moderate to large effect size between pre and post in HOT ($d$ = -0.52 to -0.80); [c]indicates moderate to large effect size between HOT and CON trials at specified target-dot numbers ($d$ = 0.52 to 0.70). *Data sets on the x-axis are staggered to prevent overlap of error bars. Each point shows Mean ± SEM.*

**Table 1. Cognitive scores and number of participants for the counting span task pre and post-surgery in CON and HOT surgeries.**

| | Counting task | | Recall Task | | | |
|---|---|---|---|---|---|---|
| | # of correct responses | | # of correct responses | | Counting span score | |
| | *PRE* | *POST* | *PRE* | *POST* | *PRE* | *POST* |
| **CON** | 45 ± 13 | 47 ± 12 | 42 ± 11 | 41 ± 11 | 5.6 ± 0.9 | 5.6 ± 1.2 |
| n | 13 | 18 | 13 | 18 | 21 | 21 |
| **HOT** | 44 ± 13 | 49 ± 9 | 39 ± 14 | 40 ± 10 | 5.3 ± 1.5 | 5.3 ± 1.0 |
| n | 18 | 18 | 18 | 18 | 18 | 18 |

All data expressed as Mean ± SD

interaction between theatre temperature and time on recall latency ($p$ = .821; Fig 2), number of correct responses (p = .828; Table 1), or overall counting span score (p = .949; Table 1).

## Manual dexterity

There was no effect of theatre temperature on manual dexterity when using the dominant hand ($p$ = .460) or the non-dominant hand ($p$ = .099). There was no interaction between theatre temperature and time on manual dexterity in either the dominant hand ($p$ = .428) or the non- dominant hand ($p$ = .949). However, when using the dominant hand there was a difference between manual dexterity pre and post-surgery ($p$ = .015), indicating an improvement over time (S3 Table).

## Perceptual responses

There was a significant effect of theatre temperature on TS ($p$ = .002), TC ($p$ < .001), and RPE ($p$ < .001), indicating that staff felt hotter, more uncomfortable, and were exerting themselves more in the heat. For all measures, scores post-surgery were significantly higher than pre-

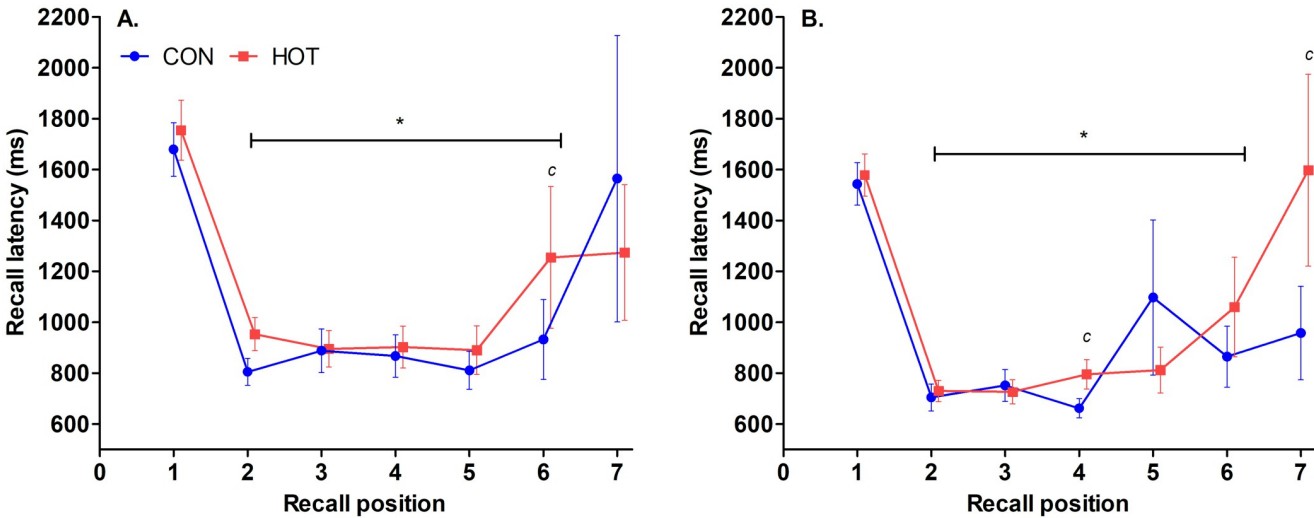

**Fig 2. Recall latencies 'pre' (A) and 'post' (B) surgery for serial recall positions 1 to 7 in CON (Pre; n = 13, Post; n = 18) and HOT (n = 18) surgeries.** *indicates response numbers that are significantly different from 1st response ($p$ < .05); $^c$ indicates moderate to large effect size between HOT and CON trials at specified set size (d = 0.74 to 0.89). *Data sets on the x-axis are staggered to prevent overlap of error bars. Data are noisy at serial recall positions 6 and 7 because only few trials had a set size > 5 (whereas all trials had serial positions 1 and 2 and many trials had positions 3–5) and because not every participant made it to a counting span of 6 or 7. Each point shows Mean ± SEM.*

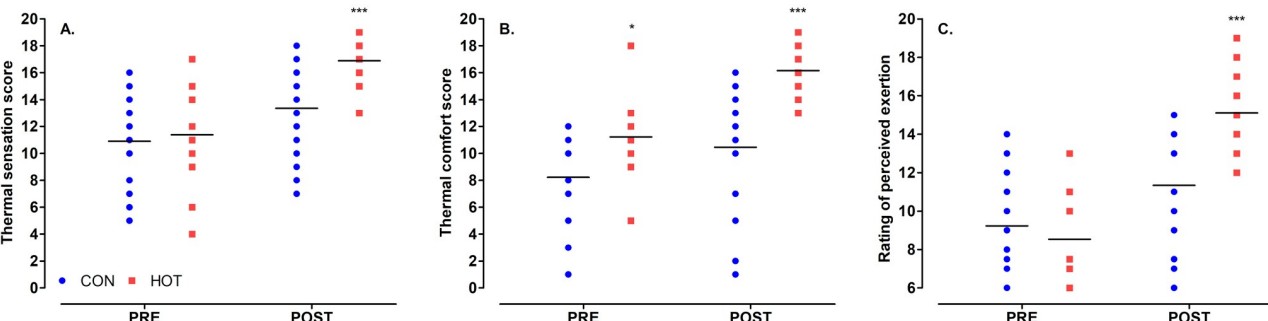

**Fig 3. Perceptual responses 'pre' and 'post' surgery in CON (n = 22) and HOT (n = 18) surgeries; thermal sensation (A), thermal comfort (B), and perceived exertion (C).** *indicates significant difference between conditions pre-surgery ($p < .05$); ***indicates significant difference between conditions post-surgery ($p < .001$). *Individual and mean data shown.*

surgery: TS ($p < .001$), TC ($p < .001$), and RPE ($p < .001$; Fig 3). There was an interaction between theatre temperature and time on TS ($p = .019$), TC ($p = .047$), and RPE ($p < .001$). The interaction supported that scores post-surgery were higher in HOT than CON for all perceptual measures ($p < .001$). It also revealed that ratings of TS and RPE were the same pre-surgery in CON and HOT ($p > .924$), but a difference in ratings of TC existed before surgery (HOT: 11 ± 3, CON: 8 ± 4; $p = .025$).

There was a significant main effect of theatre temperature on all domains of the SURG-TLX questionnaire, indicating that scores were significantly higher in the HOT compared to the CON condition (Fig 4). These domains included self-reported levels of mental demand ($p = .001$), physical demand ($p < .001$), temporal demand ($p = .007$), task complexity ($p < .001$), situational stress ($p < .001$), level of distraction ($p < .001$), and frustration ($p < .001$).

## Physiological parameters

There was a significant effect of theatre temperature on $T_{CORE}$ ($p < .001$) and HR ($p < .001$), while only $T_{CORE}$ increased over time during surgery ($p < .001$; Fig 5). There was no interaction between theatre temperature and time on $T_{CORE}$ ($p = .138$) or HR ($p = .700$).

There was a significant effect of theatre temperature on decrease in body-mass in kg ($p = .008$) and as a % of total body-mass ($p < .001$), in that the decrease in body-mass over time was greater in the HOT condition (Fig 6). The difference between rate of decrease in body mass between CON and HOT approached significance ($p = .052$). There was no effect of theatre temperature on $U_{SG}$ scores ($p = .338$); however, scores significantly increased over time ($p < .001$), indicating a greater degree of dehydration post-surgery compared to pre-surgery. There was no interaction between theatre temperature and time on $U_{SG}$ scores ($p = .138$; Fig 6). Post-surgery, $U_{SG}$ scores had a tendency to be higher in HOT compared to CON, as shown by a moderate effect size ($d = 0.50$ [-0.22, 1.15]). The number of staff members in each hydration category pre and post-surgery is provided in S4 Table.

## Discussion

To our knowledge, this is the first study to explore the effects of operating in the heat during real-time burn surgery. There were no statistically significant differences between conditions for any performance variable assessed and in general, these findings did not support our hypothesis that working in the heat would impair manual dexterity and cognitive function. Higher levels of fatigue and subjective workload found in the hot surgeries support our second

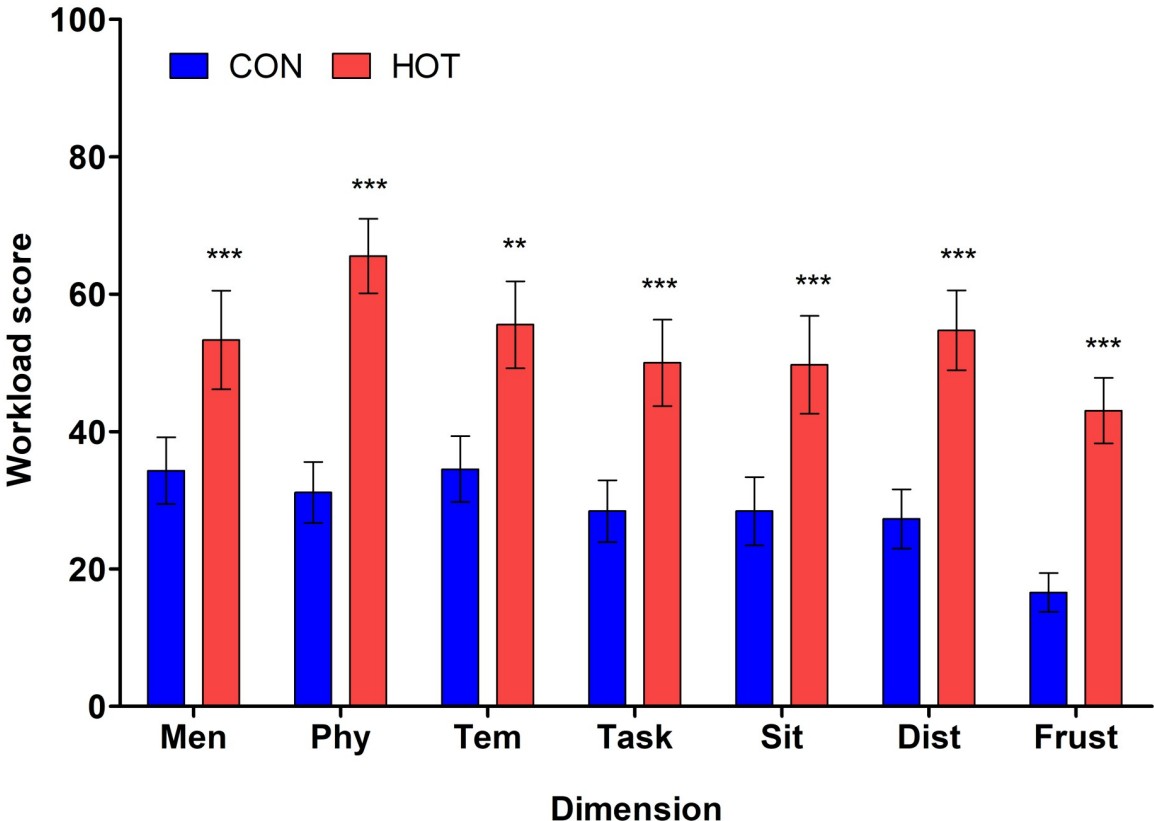

**Fig 4. Scores for each dimension of the task load index in CON (n = 22) and HOT (n = 18) surgeries; mental demand (Men), physical demand (Phy), temporal demand (Tem), task complexity (Task), situation stress (Sit), distractions (Dist), and frustration (Frust).** **indicates significant difference between conditions ($p < .01$); ***indicates significant difference between conditions ($p < .001$). *Results are presented as mean ± SEM.*

hypothesis that the heat would negatively affect perceptual responses in staff, most likely due to the higher $T_{CORE}$, HR, and fluid loss in the HOT condition, either alone or in combination.

Manual dexterity in both hands was similar between conditions. Similarly, researchers have reported no difference in dexterity scores on the O'Conner test when ambient conditions of 20 and 30°C were compared, although a significantly lower score was reported at a lower temperature of 10°C [32], possibly because cold stress, as opposed to heat stress, tends to impair manual dexterity as it decreases maximum voluntary grip strength [33]. Improvements in manual dexterity in the dominant hand over time, as found in this study, were also seen by Palejwala and colleagues [16] who attributed the improvement to various mechanisms such as decreased stiffness of muscle fibres during contraction, and reduced muscle and joint viscous resistance [34, 35].

Latency on the counting task tended to improve over time while latency for the recall task significantly improved over time in both conditions, which may be due to an increase in motor nerve conduction velocity that accompanies an increased $T_{CORE}$ [36], and an increase in arousal via activation of thermoregulatory mechanisms [37]. Scores on the counting span task, accuracy on counting and recall task, counting latency, and recall latency did not differ between conditions. Heat exposure can cause cognitive impairment, but the average $T_{CORE}$ of our staff in the heat did not exceed 38.5°C, the temperature at which cognitive tasks that require working memory tend to become impaired [38]. Cognitive function also may have

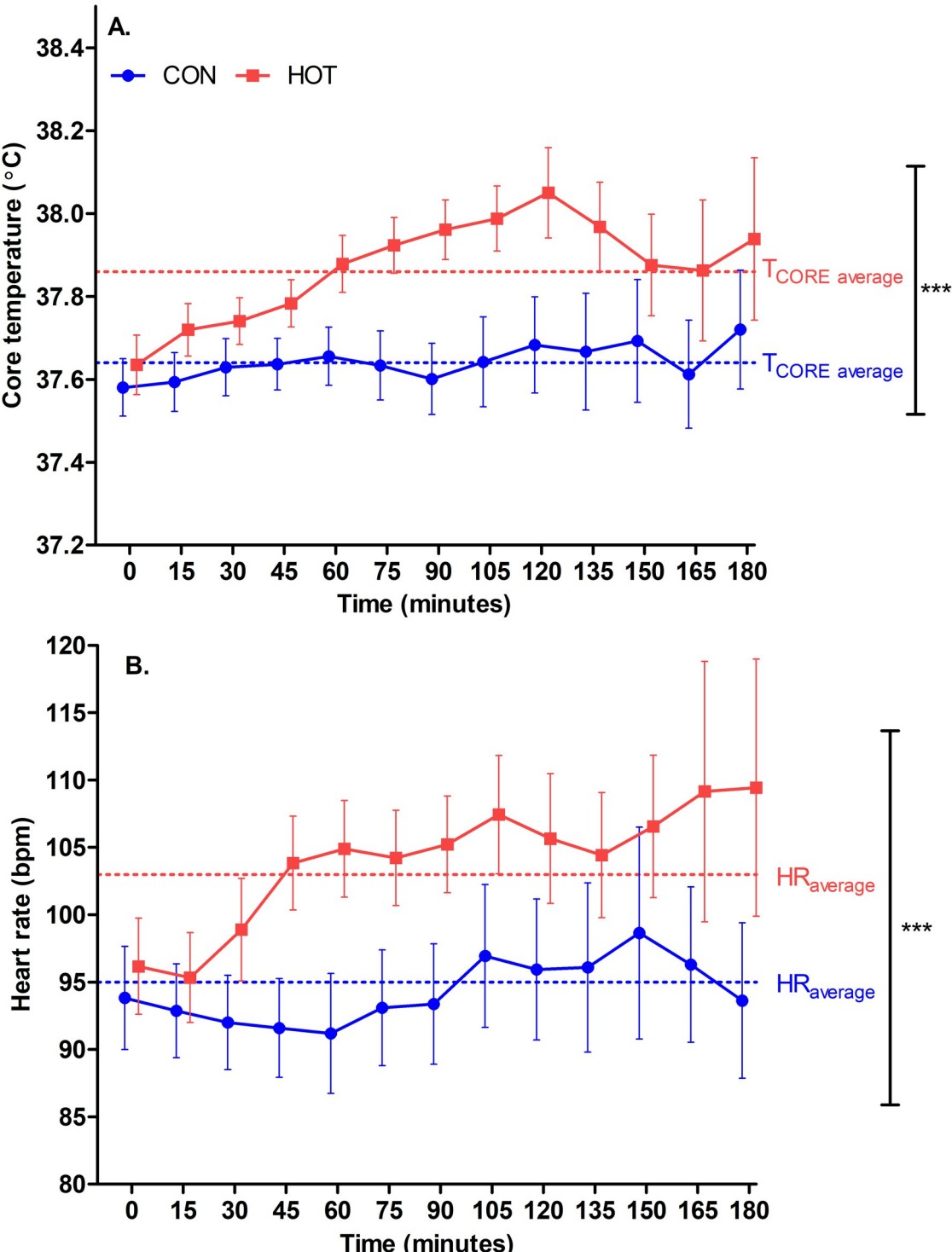

**Fig 5. Core temperature (A) and heart rate (B) responses at 15-min intervals in CON (n = 22) and HOT (n = 18) surgeries.**
***indicates significant difference between conditions ($p < .001$); *n.b Time points beyond 180 min were removed from the plots as the sample size beyond 180 min was too small ($n < 5$) to accurately represent the trend in core temperature. Data points are staggered to prevent overlap of error bars. Each point shows Mean ± SEM.*

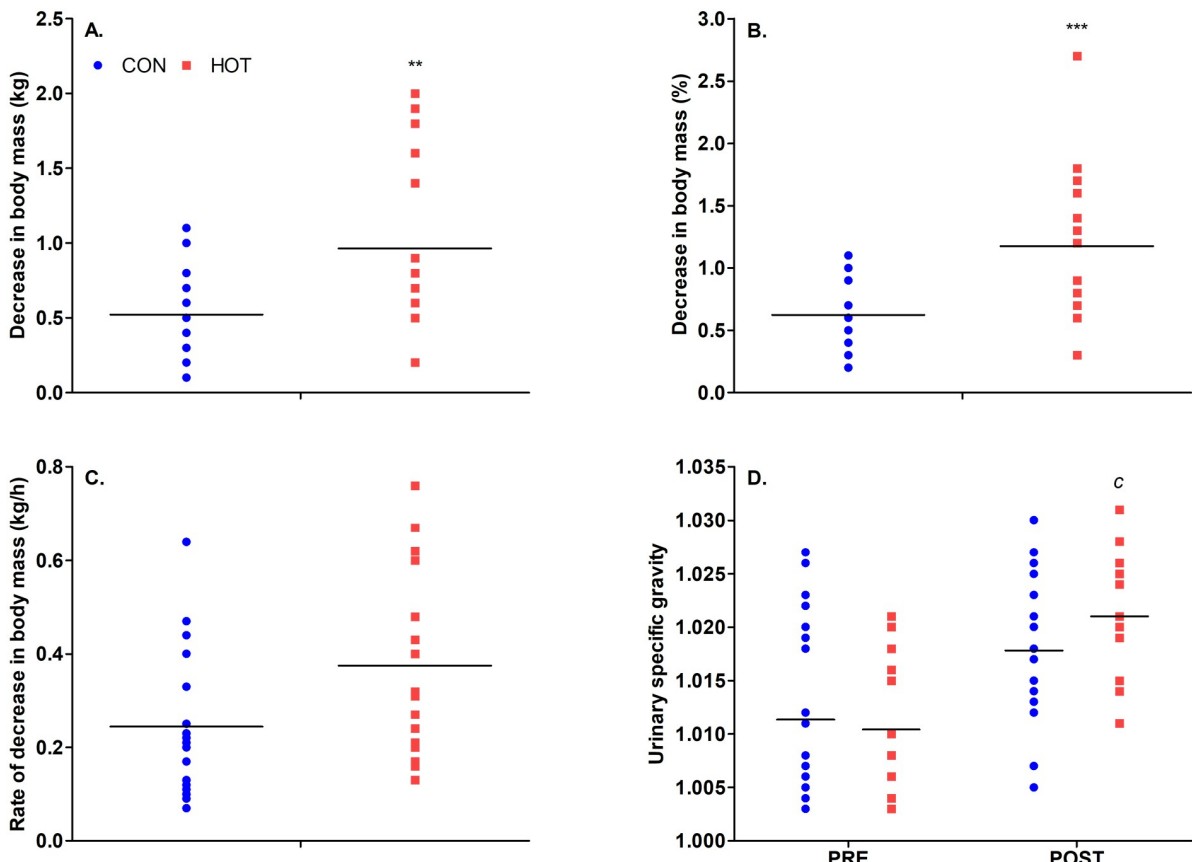

**Fig 6. Fluid loss and hydration in CON (n = 22) and HOT (n = 17) surgeries; Decrease in body mass in kg (A), decrease in body mass in % of total body mass (B), rate of decrease in body mass (C), and U$_{SG}$ scores (D).** **indicates significant difference between conditions pre-surgery ($p < .01$); ***indicates significant difference between conditions post-surgery ($p < .001$); $^c$ indicates a moderate effect size between USG scores in HOT and CON 'post' surgery ($d = 0.50$). *Individual and mean data shown.*

remained unaffected in the heat because the average % loss in body-mass was less than 2%, the critical level at which impairments to cognitive performance are commonly seen [39]. Specifically, our findings in relation to cognitive accuracy and latency are in contrast to findings of Ward and colleagues [2] who reported impaired accuracy and slowed response times in hot, surgical simulations. However, the ambient temperature in the hot simulations [2] was 34°C compared to the 30.8°C recorded in the current study. It is possible that the stronger heat stimulus in the simulations contributed to the difference, which would indicate that burn surgeries conducted in higher temperatures of up to 40°C [1], a common clinical protocol, could be of concern to the cognitive function of staff. In general, people with a high skill level, who perform tasks that are familiar or autonomous in nature, are able to withstand the effects of heat stress [40] and are therefore less susceptible to interference between stimulus and response [41, 42]. Because surgery staff are highly skilled individuals and are likely to be highly motivated, it is possible that these factors helped maintain their performance on the cognitive tasks despite increased perceptual responses to the heat. Heated surgeries of longer duration, where T$_{CORE}$ and fluid loss are likely to increase, may impair cognitive performance.

Ratings of TS, TC, and RPE were all higher post-surgery in the HOT compared to CON condition, and these findings are supported by the literature providing evidence for perceptual measures being affected by T$_{CORE}$, HR, and fluid loss [6, 43–45], all of which were significantly

higher in the HOT condition. As expected, the perceived workload was significantly higher in the HOT compared to the CON condition, which aligns with previous findings that subjective workload was greater during surgical simulations in a hot, compared to cooler environment [2]. An increase in perceived workload has been correlated with burnout, especially in the health care sector [17], which is of great importance as burnt out employees often have poor mental health and an increased risk for cardiovascular disease [18, 19]. However, it is important to note that if heated burn surgeries are not a frequent occurrence, staff may not experience the burnout-related effects of continuously working in a thermally stressful environment.

Notably, the physical demand of the surgical task, i.e., how physically fatiguing the procedure was, was higher in the HOT compared to CON theatre. Increased levels of fatigue reported during surgery could be of significant consequence in the health care industry, since fatigue is associated with an increased risk of medical errors [46, 47], carelessness among health care workers [48], and impairment to physical and mental performance during simulated medical work [49]. The level of distractions reported was also higher in the HOT compared to the CON theatre. Previously, health care workers reported that wearing PPE made them feel hot and uncomfortable at work [50], and this discomfort, coupled with the heat stress, could distract staff from their tasks. Distractions in an OT are common, but when exacerbated by heat stress, they can have a cumulative effect and possibly impact on staffs' vigilance and in turn impair operative performance [51]. In the OT, distractions and frustration can negatively affect technical performance, with staff feeling clumsy, shaky, less dexterous, and making mistakes including badly placed stitches [52]. This can have serious implications on surgical procedures and therefore patient outcomes.

## Limitations

Limitations of this study include the small sample size, and that staff were recruited from the same hospital, which introduces sampling error and reduces generalisability. Food and fluid intake as well as activities undertaken the night before surgery were not controlled for, as some staff were on call the night before, but this accurately reflects the real-world environment and job demands in health care. Exposure time in the OT was less than previous simulation time [2] in which performance differences were found, indicating that longer, real-time surgeries in the heat may lead to performance decrements. Further, the complexity of surgeries in the environmental conditions was not matched (surgeries conducted in the heat are more time sensitive in nature due to the difficulty in controlling patients body temperature [53]), which adds bias to the perception of workload, however surgery duration and TBSA was similar between conditions.

## Practical implications

The maintenance of cognitive function and manual dexterity in the heat demonstrates that burn surgery staff can maintain their working memory function and manual dexterity despite the effects of heat stress, however heat exposure can increase mental workload [13]. Long-term, continuous work with a high mental workload is correlated with cumulative fatigue and job burnout, especially in the health care sector [17], which may impact the workforce. Alleviating symptoms of heat strain in burn surgery staff should be a priority and could be achieved by taking small breaks during surgery, using underbody warming devices for the patient (warming mattresses) as opposed to heating the OT, or cooling technologies for staff. For example, head cooling caps [54] and cooling vests [55] have been found to lower perceptual heat strain and may be able to do the same for surgery staff working in the heat, thereby lowering their mental workload.

## Conclusions

This study showed that cognitive function and manual dexterity was maintained while operating in the heat, however subjective workload and fatigue increased, possibly due to heat strain. Our results suggest that it would be beneficial to consider fatigue/the physical demand of tasks and mental workload in the work design for major burns (heated surgeries). Future research should 1. build on this study and assess cumulative fatigue in burn surgery staff over a longer period of time, and 2. find the optimal temperature for burns OTs in which performance can be maintained while considering factors to lower the level of heat strain and workload of staff.

## Supporting information

**S1 Table. Participant numbers and demographic information for CON and HOT surgery conditions (mean ± SD).**
(PDF)

**S2 Table. Assignment of participants to surgeries (*Sx*) in both environmental conditions.**
(PDF)

**S3 Table. Scores for the Purdue pegboard task pre and post-surgery in CON (n = 22) and HOT (n = 18) surgeries for the dominant and non-dominant hand.** All data expressed as Mean ± SD.
(PDF)

**S4 Table. Number and percentage of participants *n (%)* in each hydration status category 'pre' and 'post' surgery, in CON and HOT surgeries.**
(PDF)

**S1 File. Core temperature and heart rate data.**
(XLSX)

**S2 File. USG and sweat loss data.**
(XLSX)

**S3 File. Pegboard data.**
(XLSX)

**S4 File. Cognitive scores data.**
(XLSX)

**S5 File. Cognitive latency scores.**
(XLSX)

**S6 File. Perceptual responses data.**
(XLSX)

**S7 File. Temperature and humidity data.**
(XLSX)

## Acknowledgments

The authors thank the staff at Fiona Stanley Hospital for their participation, and statistician Martin Firth for his consultation.

## Author Contributions

**Conceptualization:** Karen E. Wallman, Shane Maloney, Grant J. Landers, Ullrich K. H. Ecker, Fiona M. Wood.

**Data curation:** Zehra Palejwala.

**Formal analysis:** Zehra Palejwala, Ullrich K. H. Ecker.

**Funding acquisition:** Zehra Palejwala, Fiona M. Wood.

**Investigation:** Zehra Palejwala.

**Methodology:** Zehra Palejwala, Karen E. Wallman, Shane Maloney, Grant J. Landers, Ullrich K. H. Ecker, Fiona M. Wood.

**Project administration:** Zehra Palejwala.

**Resources:** Karen E. Wallman, Shane Maloney, Grant J. Landers, Fiona M. Wood.

**Software:** Ullrich K. H. Ecker.

**Supervision:** Karen E. Wallman, Shane Maloney, Grant J. Landers, Fiona M. Wood.

**Validation:** Ullrich K. H. Ecker.

**Visualization:** Shane Maloney, Grant J. Landers, Ullrich K. H. Ecker, Mark W. Fear.

**Writing – original draft:** Zehra Palejwala.

**Writing – review & editing:** Zehra Palejwala, Karen E. Wallman, Shane Maloney, Grant J. Landers, Ullrich K. H. Ecker, Mark W. Fear, Fiona M. Wood.

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
