## [Decision Letter · Decision Letter 0]

13 Apr 2023

PONE-D-23-05820Higher operating theatre temperature during burn surgery increases subjective workload and fatigue as a result of heat strainPLOS ONE

Dear Dr. Zehra Palejwala,

Thank you for submitting your manuscript to PLOS ONE. After careful consideration, we feel that it has merit but does not fully meet PLOS ONE’s publication criteria as it currently stands. Therefore, we invite you to submit a revised version of the manuscript that addresses the points raised during the review process.

Three experts in the field reviewed the present study. The manuscript provides novel and exciting data and may be suitable for publication in PLOS One. However, it can still be improved (some issues should be clarified) before it is ready for publication. Although the reviewers did not indicate flaws or methodological issues that cannot be corrected in the revised manuscript, they have presented suggestions/corrections to all sections: from the title to the conclusions. Please take every comment very seriously and resubmit a better manuscript. This academic Editor is looking forward to receiving a revised and improved version of the manuscript for further analysis.

We look forward to receiving your revised manuscript.

Kind regards,

Samuel Penna Wanner, Ph.D.

Academic Editor

PLOS ONE

Journal Requirements:

Additional Editor Comments (if provided):

I have nothing to add to the reviewers' comments.

Reviewers' comments:

Reviewer's Responses to Questions

**Comments to the Author**

1. Is the manuscript technically sound, and do the data support the conclusions?

Reviewer #1: Yes

Reviewer #2: Yes

Reviewer #3: Yes

2. Has the statistical analysis been performed appropriately and rigorously? 

Reviewer #1: Yes

Reviewer #2: Yes

Reviewer #3: Yes

3. Have the authors made all data underlying the findings in their manuscript fully available?

Reviewer #1: Yes

Reviewer #2: No

Reviewer #3: No

4. Is the manuscript presented in an intelligible fashion and written in standard English?

Reviewer #1: Yes

Reviewer #2: Yes

Reviewer #3: Yes

5. Review Comments to the Author

Reviewer #1: The authors completed attractive work overall, with the aims and methods section being interesting. My comments are all minimal. However, I would like them to answer them. Thanks for the invitation to review this interesting work.

Reviewer #2: Palejwala and colleagues aimed to assess the real-time impact of heat during burn surgeries on staff’s cognitive performance, manual dexterity, and perceptual measures (workload, thermal sensation, thermal comfort, perceived exertion, and fatigue) and physiological parameters (core temperature, heart rate, fluid loss, and dehydration).

The results are quite relevant, mainly because (according to the authors) the impact of heat on burn surgical teams has not been previously measured in a real-world context. There are some questions that should be elucidated and corrected.

I have listed minor points aimed at improving the quality of the manuscript as follows:

1. The authors should detail more about “All staff provided information on previous heat exposure (i.e., recent holidays in hot climates and weekly, surgical heat exposure) and none were determined to be acclimatized/acclimated”. This simple information is not able to refute a hypothesis of acclimatized/acclimated participants. For example, aerobic fitness can acclimatize/acclimate humans. In this sense, is fundamental to expose a table with the sample characteristics (age, weight, level of aerobic fitness or amount of weekly physical activity, experience time as a surgeon, etc.).

2. “An increase in perceived workload is correlated with burnout, especially in the health care sector [14]”. Are acutes perceived workloads able to cause burnout?

3. Why do you use “hot” and “thermoneutral”, and not “heat”, “warm” and “temperate” environments?

4. Why were three participants only tested in HOT?

5. How did you measure the body-mass? Insert this information in the methods section.

6. Page 3, line 65. Please Insert reference.

7. Page 5, line 129. Please Insert a reference for this method.

Reviewer #3: Dr. Zehra Palejwala and colleagues assessed the impact of environmental heat stress during burn surgeries on staff’s cognitive performance, manual dexterity, perceptual measures (workload, thermal sensation, thermal comfort, perceived exertion, and fatigue), and physiological parameters (body core temperature, heart rate, fluid loss, and dehydration). The authors reported that the performance variables were unaffected in the heat. However, they observed higher levels of fatigue and subjective workload in hot surgeries, suggesting that environmental heat stress negatively affected perceptual responses in staff, most likely due to the higher body core temperature, heart rate, and fluid loss in the hot condition, either alone or in combination.

The topic addressed in the manuscript interests healthcare workers and researchers investigating the quality of life/burnout in this population. The current study advances previous literature because it investigates the impact of environmental heat stress on performance and psychophysiological responses in real-world (not simulated) contexts. The manuscript is well written and has adequate size. The methods are adequate to investigate the research problem. The results supporting the conclusions are presented in good figures and tables. Despite these strong points, the authors should address some issues before the manuscript is ready for publication.

Please see my suggestions to improve the manuscript below.

Major points

1. Title, lines 2 and 3. The authors indicated increased perceived workload and fatigue resulted from augmented heat strain. Because of the experimental design, the statement indicating a cause-effect relationship between heat strain and perceived workload/fatigue seems too strong. Moreover, the title should indicate whether the changes were assessed in patients or surgeons. Please consider the following alternative title: "Higher operating theatre temperature during burn surgery increases the surgical staff's physiological heat strain and subjective workload and fatigue".

2. The authors should be less emphatic in their conclusions in the abstract and regular text. For example, the experiments in the heat were conducted at the lower end of the recommended ambient temperature range for major burn surgeries (i.e., 30°C to 40°C). Therefore, the authors should exercise caution to state that "operating in the heat is a safe approach for patient treatment" (line 54) because this may not be true for surgeries at 35°C.

2.1. Abstract, lines 54 to 55. The information about job burnout was poorly contextualized in this section and appeared first in conclusion. To amend this issue, the authors may want to mention that job burnout is positively correlated with the perceived workload.

2.2. Lines 381 and 382. Please consider replacing “heat stress” with “heat strain” and insert the word “possibly” before “due to”.

3. Did the patients become hypothermic during surgeries under control conditions? This information should be inserted in the revised manuscript to ensure researchers took proper ethical care.

4. Did the different scales used in the current study provide overlapping information? For example, what are the differences between data provided by Borg's scale, the 20-point visual-analogue scale for determining "physical fatigue", and the physical demand domain of the perceived workload scale?

4.1. Results, lines 237 and 239. The fact that overlapping information exists was evidenced at the end of the following sentence: “There was a significant effect of theatre temperature on TS (p = .002), TC (p < .001), RPE (p < .001), and fatigue (p < .001), indicating that staff felt hotter, more uncomfortable, more fatigued, and were exerting themselves more in the heat (Fig 3)”.

4.2. Panels C and D in Figure 3 are very similar, thus basically providing the same information.

4.3. If these scales provide overlapping information, please consider removing one from the manuscript. For example, I would suggest removing the 20-point visual-analogue scale for determining "physical fatigue", because it was initially used to measure mental fatigue (lines 147 and 148).

5. While reading the Results section, I noticed that the authors could clearly describe the main effect of time or the main effect of ambient temperature. However, I missed the information about the time x ambient temperature interactions. This issue is evident when analyzing data presented in figures and Table 1. Therefore, please include the information about interactions in the revised manuscript.

6. Although the figures were well elaborated, they can still be improved in several ways. First, most font sizes should be increased, especially in Figures 3, 4, and 6. Second, the white space between panels should be reduced in all figures. For example, the letters indicating the panels can be placed closer to or even inside these panels. Third, PLoS One is a journal published online, and it does not charge an additional tax to publish color figures. The manuscript will benefit from using blue symbols for the control condition and red symbols for the hot condition.

6.1. Figure 4. The authors may want to present a panel with the overall score for the task load index (or workload), thus reducing the white space in this figure.

Minor points

1. Abstract, line 38. When mentioned for the first time, it may be interesting to emphasize that authors are referring to the term "body core temperature". After that, writing only "core temperature" or using the corresponding abbreviation is okay. This suggestion is also valid for the regular text (line 126).

2. Abstract, line 46. Please define the meaning of the “RH” abbreviation before its first appearance.

3. Introduction, line 82. Please consider adding “not simulated” in the following sentence to improve clarity: “measured in a real-world (not simulated) context”.

4. Methods. Comparisons in lines 104 to 108 should be moved to the Results section. If the authors decide to maintain this information in the methods (although not ideal), please explain the meaning of the data (i.e., mean ± SEM) and include the statistical test used to generate the p-values reported.

5. Methods, line 107. The average total body surface area was less than 20% (i.e., 8 ± 13%) during surgery in control conditions. This means that surgery was not necessarily a major burn surgery. Please amend the sentence in line 384.

6. Results, lines 190 to 193. To improve the description, the authors should mention the menstrual phase of the following two women surgeons: “… one was only tested once, and one was in the same menstrual cycle phase during testing sessions”.

7. Table 1, lines 209 to 211. Please consider replacing “49 ± 9.0” with “49 ± 9”.

8. Discussion, line 306. Please indicate the number of the reference instead of the publication year.

9. Discussion, line 376. Please consider citing the following reference – doi: 10.1080/23328940.2020.1868386 – or any other relevant reference.

6. PLOS authors have the option to publish the peer review history of their article (what does this mean?). If published, this will include your full peer review and any attached files.

Reviewer #1: No

Reviewer #2: **Yes: **Alexandre SR Hudson

Reviewer #3: No

---

## [Author Response · Author response to Decision Letter 0]

29 Apr 2023

General comments for revision submission 

Please ensure that your manuscript meets PLOS ONE's style requirements, including those for file naming. The PLOS ONE style templates can be found at https://journals.plos.org/plosone/s/file?id=wjVg/PLOSOne_formatting_sample_main_body.pdf and 

Response:

Submission guidelines including style requirements, file naming and author affiliations have been thoroughly checked and we believe the manuscript adheres to all guidelines. 

Please provide additional details regarding participant consent. In the ethics statement in the Methods and online submission information, please ensure that you have specified what type you obtained (for instance, written or verbal, and if verbal, how it was documented and witnessed). If your study included minors, state whether you obtained consent from parents or guardians. If the need for consent was waived by the ethics committee, please include this information.

Response:

The ethics statement on page4, from lines 92-96 of the materials and methods section has been amended and now reads: “All staff gave written consent for participation in this study and patients gave either written or verbal consent (written consent was not able to be provided by all patients because of the nature and location of the burn injury). Verbal consent was witnessed by a member of the surgical burns team who then signed and dated the consent form, attesting that the requirements for informed consent were satisfied. Ethical approval was granted by the Human Research Ethics Committee of the University of Western Australia (2020/ET000239) and the South Metropolitan Health Service Human Research Ethics Committee (PRN RGS0000004250).” 

Response:

The appropriate text has been added to the ‘ethics statement’ field of the submission form.

Reviewer # 1 

GENERAL COMMENTS 

The authors completed attractive work overall, with the aims and methods section being interesting. My comments are all minimal.

Response:

We thank the reviewer for their positive appraisal and valuable comments. We have done our best to address each comment below and revise the manuscript accordingly. Amended sentences appear highlighted in red in the revised manuscript.

SPECIFIC COMMENTS

I see cognitive performance as a complex phenomenon. Thus, analyses involving three cognitive domains provide limited explanations for something as complex as the regulation of cognitive performance. If you don't have a justification, consider using executive function performance and not cognitive performance.

Response:

We agree that working memory, assessed in our study, is a cognitive domain rather than something that completely encompasses cognitive performance, however some of the measures from the counting-span task are simple counting scores, latencies, and counting dots, which do not measure executive function in a meaningful way. Thus, we have used the term “cognitive function” as a better umbrella term for counting and serial recall performance (while making no claim that cognitive function was assessed in an exhaustive manner). Additional information has been inserted in the manuscript to provide clarity in the cognitive task used and what it assessed. The manuscript on lines 176-178 now reads: “As such, measures included those that relate to basic counting performance as well as those assessing working memory capacity; for this reason, the term ‘cognitive function’ has been used to refer to these measures collectively.”

Line 156. Working memory is not a component of executive function? Please be more explicit.

Response:

The sentence on line 164 has been modified accordingly, avoiding the term executive function, and it now reads “The function of working memory – a core component of executive capacity – was measured by the counting span task (millisecond software) on a laptop.”

Line 99. I am not native to an English-speaking country, but if there is no specific reason, consider using CON and not CONTROL.

Response: 

We have changed the term “CONTROL” to “CON” throughout the manuscript. These changes have occurred on lines 42, 48, 91, 102, 105, 106, 108, 201, 203, 204, 226 (table legend & table), 264, 265, 300, 304, 354, 366, 371. The term “CONTROL” was also changed to “CON” in all figures, figure legends, and supplementary files.

Line 59. Workload and fatigue already appear in the title. Consider defining other keywords. Perhaps this increases the chances of your article being found in a database.

Response:

Thank you for this observation. On line 55, the keywords “workload” and “fatigue” have now been replaced with alternative terms; “surgery task load index” and “physical demand.”

Line 109. Considering the results of this study, I would suggest head cooling during burn surgery. Perhaps you should know about a recent work on the topic (doi: 10.1111/sms.13985).

Response:

We agree and thank you for providing the resource above. In the practical implications section, where there is mention of cooling technologies for staff, the use of head cooling and a cooling vest (as per an additional comment from a reviewer) has been suggested. On lines 399-404, the section now reads: “Alleviating symptoms of heat strain in burn surgery staff should be a priority and could be achieved by taking small breaks during surgery, using underbody warming devices for the patient (warming mattresses) as opposed to heating the OT, or cooling technologies for staff. For example, head cooling caps [54] and cooling vests [55] have been found to lower perceptual heat strain and may be able to do the same for surgery staff working in the heat, thereby lowering their mental workload”. 

Line 129. Consider using TCORE. I think it improves visualization.

Response:

Throughout the manuscript the term Tcore has been altered to read TCORE. These changes have occurred on lines 136, 139, 145, 287, 288, 289, 321, 333, 336, 352, and 356.

Line 136. Why was water not provided ad libitum for the medical team?

Response:

To clarify, water was not provided ab libitum for the medical team as fluids were not permitted in the operating theatre. However, if the team desired water, they were able to ‘scrub out’, leave the operating theatre and consume fluids. It is not common practice to exit the theatre during a procedure and all bar one chose to refrain from drinking to minimise bathroom breaks and avoid having to scrub out.

Line 139. Just a suggestion. You could have corrected the BM pre- and post-test for the volume of water ingested (Please note the question before), the clothes you wore, and the weight of the experimental instruments. 

Response:

Thank you for your comment. Only one participant in one trial consumed fluids during surgery and this amount was recorded and accounted for in the body mass calculations. In no other trials were any fluids consumed after the initial body mass measurement was taken and before the final one was taken. On lines 140-142, within the protocol section of the methods, it is stated that measurements were taken with participants in the nude, in private, therefore the weight of clothes was not considered in the calculation. The same experimental instruments were used pre- and post- surgery for everyone to account for the weight of experimental instruments.

Line 176. Why didn't you use a more up-to-date version of R Studio for Windows (e.g., version 2022.07.2)? I think this may have reflected in the quality of the figures. Use colors and the ggplot2 package.

Response:

All figures were created using GraphPad Prism 5 for windows, version 5.00, 2007. Initially, all figures were saved as resolution 300 dpi but have now been exported as resolution 600 dpi to improve the quality. 

Line 182. I'm not a statistician, but I was concerned about Tukey's use of post hoc. I think some variables have a coefficient of variation (CV) ≥ 15%. Tukey's post hoc is very rigorous; consider using a less rigorous one for variables with a high CV. I consider in my post hoc choices the number of treatments and the CV of the variable. Thus, I minimize the chances of type I and II errors (DOI: 10.1177/0013164488483001; ISBN: 9788587144522).

Table. Determining the post-hoc test

Variable instability CV Number 

of treatments Post hoc test

Low ≤15% ≤5 Tukey, SNK or Scheffé

Low ≤15% >5 SNK or Tukey

Medium 15 a 30% ≤4 Student's t test

Medium 15 a 30% ≥5 SNK or Duncan

High ≥30% ≤4 Student's t test

High ≥30% >4 Duncan

Response: 

Thank you for providing the resource above. We have spoken to a statistician at the university who has informed us that the “Tukey” post hoc method is an appropriate test to use for our data analysis. For this reason, we have left the manuscript as is, with the Tukey post-hoc test.

Line 189. If you don't have a specific reason, consider using relative humidity instead of Relative Humidity.

Response:

On line 43 in the abstract, the term “Relative Humidity” has now been corrected to “relative humidity”. Following an additional observation from one of the reviewers, the term relative humidity was not defined prior to its abbreviation in the abstract of the manuscript and so the term “relative humidity” has been inserted in the abstract and removed from the results section, where it was first written as “Relative Humidity”. 

Line 198. If you don't have a cause in mind, think about using Figure rather than Fig.

Response:

The term “Fig” has been used throughout the manuscript, rather than “Figure”, in accordance with the PLOS One manuscript body and formatting guidelines.

Lines 183, 207, and 228. Are all results shown expressed as mean ± SD or mean ± SEM? 

Response:

The data in the written text and tables are presented as mean ± SD throughout the manuscript however in the figures it is presented as mean ± SEM due to standard deviations being large and overlapping figure data points. In all figure legends we have noted whether or not data are presented as mean ± SD, mean ± SEM, or whether all individua data points are plotted. In the statistical analysis section of ‘Materials and methods’ we have provided clarity on line 194-195 by noting that “All results presented within the written text and tables are expressed as mean ± SD and all figures are presented as individual data points or mean ± SEM”. 

Line 279. I am aware of the study by Casa et al. (2000); however, you could be a little more cautious in stating that individuals have significant dehydration (consider using hypohydration). Measurement of urine specific gravity may not reflect plasma osmolality, the most efficient measure to assess hydration status. Perhaps you should know and consider the article by Pereira et al. (2017) (DOI: 10.23736/S0022-4707.16.06836-5). 

Response:

Thank you for providing the resource above. Within the materials and methods section, after the mention of USG measurement and classifications, a sentence citing the reference above has been added, to exercise caution in the way hydration is classified. The section from lines 132-135 now reads: “Values obtained for USG were classified as ‘well hydrated’ <1.010, ‘minimal dehydration’=1.010-1.020, ‘significant dehydration’=1.021-1.030 and ‘serious dehydration’ >1.030 [22]. It is important to note that measurement of USG may not reflect plasma osmolality [23], the most efficient measure to assess hydration status, and so the classifications provided may not be accurate in illustrating the extent of hypohydration.

In supplemental table 4, which provides the number of staff members in each hydration category, a footnote reading “It is important to note that measurement of USG may not reflect plasma osmolality, the most efficient measure to assess hydration status, and so the classifications provided may not be accurate in illustrating the extent of hypohydration” has been added.

Lines 394 to 532. Important current references in the area of thermoregulation and areas similar to this study were not presented. You only presented three references from the last three years. Please consider citing these: DOI: 10.1371/journal.pone.0274584; DOI: 10.1038/s41572-021-00334-6; DOI: 10.1177/1553350620934931; DOI: 10.1308/rcsann.2020.7001.

Response:

Thank you for providing the resources above. The following references, DOI: 10.1038/s41572-021-00334-6 and 10.1177/1553350620934931, have been cited in the revised manuscript in the introduction. On lines 61-65, the section now reads: “This can improve patient outcomes; however patient outcomes also depend on the cognitive function [2], manual dexterity/technical skills [3], and fatigue levels of surgical teams [4]. Heat exposure can lead to heat and cardiovascular strain [5], and dehydration if fluids are not adequately replaced, all of which can impair physical and cognitive function [6-9]”. 

Reviewer # 2 

GENERAL COMMENTS 

Palejwala and colleagues aimed to assess the real-time impact of heat during burn surgeries on staff’s cognitive performance, manual dexterity, and perceptual measures (workload, thermal sensation, thermal comfort, perceived exertion, and fatigue) and physiological parameters (core temperature, heart rate, fluid loss, and dehydration). The results are quite relevant, mainly because (according to the authors) the impact of heat on burn surgical teams has not been previously measured in a real-world context. There are some questions that should be elucidated and corrected

Response:

We thank the reviewer for their positive appraisal and valuable comments. We have done our best to address each comment below and revise the manuscript accordingly. Amended sentences appear highlighted in red in the revised manuscript.

SPECIFIC COMMENTS

The authors should detail more about “All staff provided information on previous heat exposure (i.e., recent holidays in hot climates and weekly, surgical heat exposure) and none were determined to be acclimatized/acclimated”. This simple information is not able to refute a hypothesis of acclimatized/acclimated participants. For example, aerobic fitness can acclimatize/acclimate humans. In this sense, is fundamental to expose a table with the sample characteristics (age, weight, level of aerobic fitness or amount of weekly physical activity, experience time as a surgeon, etc.)

Response:

We agree with this comment and have provided additional information in the revised manuscript to support the hypothesis that staff were not acclimatised/acclimated. Data were collected in the winter months when outdoor ambient temperatures reached a maximum of 20°C, specifically to minimise the possibility of acclimatisation. This has now been clarified in the manuscript on lines 90-91, under the subheading ‘participants’ in the ‘materials and methods’, which now reads: “Surgical staff from a burns department were recruited in the winter (June - October 2021; when average maximum ambient temperature was 20°C) to minimise the possibility of acclimatisation/acclimation, for testing in CON and HOT conditions”.

In the ‘familiarisation’ section of ‘materials and methods’ additional information regarding weekly heat exposure has been provided. Lines 115-124 now read: “All staff provided information on weekly heat exposure and physical activity via the International Physical Activity Questionnaire [21], to determine acclimatisation/acclimation status. The average, maximum, outdoor temperature during the testing period was 20°C and no staff had travelled to a warmer climate in recent months prior to testing. Within the four months of testing, staff were exposed to a heated OT on 5 occurrences, with the average duration of exposure being 158 min, equating to a total average of 50 min per week. The average amount of recreational physical activity (not including job-related, indoor physical activity i.e. walking within the hospital) was 5 hours weekly, with most reporting light, as opposed to moderate and high intensity physical activity. Thus, no staff were determined to be acclimatised/acclimated”. 

“An increase in perceived workload is correlated with burnout, especially in the health care sector [14]”. Are acutes perceived workloads able to cause burnout? 

Response:

Acute periods of a high workload in isolation would not be sufficient to cause burnout, but chronic exposure to these acute periods are what accumulate and cause burnout (doi: 10.2147/LRA.S240564). Long-term, continuous work with a high mental workload is correlated with job burnout, specifically in the health care sector . 

Why do you use “hot” and “thermoneutral”, and not “heat”, “warm” and “temperate” environments?

Response: 

Thank you for your question. In the most recent journal articles (DOI: 10.1097/sla.0000000000004598 and https://doi.org/10.1371/journal.pone.0222923) assessing heat strain in burn surgery staff, the term “hot” for the operating theatre was used and the environmental conditions in this study were similar to those in the aforementioned articles. Thus, the term “hot” was used to maintain consistency with the literature. We chose to use the term “thermoneutral” as the temperature in our control condition fell within the temperatures classified as the thermoneutral ambient temperature range for someone who is lightly clothed, which is ~21.5-25°C (DOI: 10.4161/temp.29702). 

Why were three participants only tested in HOT?

Response:

The three participants who were tested in HOT only were initially tested in the HOT theatre and were to be tested in the CONTROL condition however rising COVID cases, and subsequent snap lockdowns, prevented the research team from entering the theatre to collect CONTROL data for said individuals. As such, their data were still included in the analysis and a mixed model code was used to account for the unmatched data.

How did you measure the body-mass? Insert this information in the methods section.

Response

The required information has now been added to the methods section, within the protocol write up on lines 141-142. The sentence now reads “In private, nude body mass was measured to the nearest 0.1 kg using a digital platform scale (details provided above)”, as the details of the weighing scales were included in the section titled “familiarisation session.

Page 3, line 65. Please Insert reference.

Response:

Thank you for this observation. References for the statement “patient outcomes also depend on the cognitive function, manual dexterity and fatigue levels of surgical teams” have been inserted and the statement from lines 61-63 now reads “patient outcomes also depend on the cognitive function [2], manual dexterity/technical skills [3], and fatigue levels of surgical teams [4].”

Page 5, line 129. Please Insert a reference for this method.

Response:

Thank you for this observation. On line 139, a reference has been added at the end of the statement and the sentence now reads “Four to eight hours prior to surgery, staff ingested a pill that objectively measured TCORE [24]”.

Reviewer # 3 

GENERAL COMMENTS

Dr. Zehra Palejwala and colleagues assessed the impact of environmental heat stress during burn surgeries on staff’s cognitive performance, manual dexterity, perceptual measures (workload, thermal sensation, thermal comfort, perceived exertion, and fatigue), and physiological parameters (body core temperature, heart rate, fluid loss, and dehydration). The authors reported that the performance variables were unaffected in the heat. However, they observed higher levels of fatigue and subjective workload in hot surgeries, suggesting that environmental heat stress negatively affected perceptual responses in staff, most likely due to the higher body core temperature, heart rate, and fluid loss in the hot condition, either alone or in combination. The topic addressed in the manuscript interests healthcare workers and researchers investigating the quality of life/burnout in this population. The current study advances previous literature because it investigates the impact of environmental heat stress on performance and psychophysiological responses in real-world (not simulated) contexts. The manuscript is well written and has adequate size. The methods are adequate to investigate the research problem. The results supporting the conclusions are presented in good figures and tables. Despite these strong points, the authors should address some issues before the manuscript is ready for publication.

Response:

We thank the reviewer for their positive appraisal and valuable comments. We have done our best to address each comment below and revise the manuscript accordingly. Amended sentences appear highlighted in red in the revised manuscript.

MAJOR COMMENTS 

Title, lines 2 and 3. The authors indicated increased perceived workload and fatigue resulted from augmented heat strain. Because of the experimental design, the statement indicating a cause-effect relationship between heat strain and perceived workload/fatigue seems too strong. Moreover, the title should indicate whether the changes were assessed in patients or surgeons. Please consider the following alternative title: "Higher operating theatre temperature during burn surgery increases the surgical staff's physiological heat strain and subjective workload and fatigue".

Response:

Thank you for your suggestion. We agree that the manuscript title would be improved with minor revisions and so the title has been altered to read “Higher operating theatre temperature during burn surgery increases physiological heat strain, subjective workload, and fatigue of surgical staff.”

The authors should be less emphatic in their conclusions in the abstract and regular text. For example, the experiments in the heat were conducted at the lower end of the recommended ambient temperature range for major burn surgeries (i.e., 30°C to 40°C). Therefore, the authors should exercise caution to state that "operating in the heat is a safe approach for patient treatment" (line 54) because this may not be true for surgeries at 35°C.

Response:

In the abstract, on line 50, the statement “suggesting that operating in the heat is a safe approch for patient treatment” has been altered to exercise caution and now reads “suggesting that operating in approximately 30°C heat is a safe approach for patient treatment”.

Abstract, lines 54 to 55. The information about job burnout was poorly contextualized in this section and appeared first in conclusion. To amend this issue, the authors may want to mention that job burnout is positively correlated with the perceived workload.

Response: 

On line 51, The conclusion section of the abstract has been amended to contextualise the statement regarding burnout. The sentence now reads: “However, job burnout, which is positively correlated with perceived workoad, and the impact of cumulative fatigue on the mental health of surgery staff, must be considered in the context of supporting an effective health workforce”.

Lines 381 and 382. Please consider replacing “heat stress” with “heat strain” and insert the word “possibly” before “due to”.

Response:

In the conclusion, on lines 408- 409, the statement “due to heat stress” has been revised to “possibly due to heat strain” and the manuscript now reads “This study showed that cognitive function and manual dexterity was maintained while operating in the heat, however subjective workload and fatigue/the physical demand of tasks increased, possibly due to heat strain”

Did the patients become hypothermic during surgeries under control conditions? This information should be inserted in the revised manuscript to ensure researchers took proper ethical care.

Response:

Thank you for your question. After consulting with the surgical team, we have been informed that no patients became hypothermic during surgery in either environmental condition. In the section on experimental design, following surgery information, a statement from lines 107-108 that reads “No patients became hypothermic during surgery in either the CON or HOT condition” has been added. 

Did the different scales used in the current study provide overlapping information? For example, what are the differences between data provided by Borg's scale, the 20-point visual-analogue scale for determining "physical fatigue", and the physical demand domain of the perceived workload scale?

Response:

Thank you for this question. To clarify, Borg’s scale for rating of perceived exertion is different from physical demand, which is answered by the SURG-TLX questionnaire. Borg’s scale provides a measure of how hard it feels that the body is working and encompasses effort, exertion, breathlessness as well as fatigue, to give an overall estimate of intensity of physical activity whereas the SURG-TLX domain of physical demand provides information on how physically fatiguing an individual feels a procedure or task was. The 20-point visual analogue scale however provides overlapping information with the physical demand domain of the SURG-TLX questionnaire and thus has been removed from the revised manuscript

Results, lines 237 and 239. The fact that overlapping information exists was evidenced at the end of the following sentence: “There was a significant effect of theatre temperature on TS (p = .002), TC (p < .001), RPE (p < .001), and fatigue (p < .001), indicating that staff felt hotter, more uncomfortable, more fatigued, and were exerting themselves more in the heat (Fig 3)”.

Response: 

Thermal sensation and thermal comfort, although similar, provide different information. For example, thermal sensation relates to information from the skin surface where changes are dependent on the external environment whereas thermal comfort refers to the state of mind of an individual when expressing satisfaction or dissatisfaction with the surrounding environment. Both measures have been used in conjunction in previous studies (https://doi.org/10.1016/j.jobe.2017.02.004). As noted above, fatigue and perceived exertion also provide different information and complement each other in describing how an individual feels. A recent article (https://doi.org/10.3389/fphys.2021.735565) that discusses fatigue and perceived exertion confirm that they are related but are independent measures. The article states that “Although RPE [is] considered a measure of exercise intensity, recent studies suggest that [it] could be affected by other factors, i.e., duration of the session or fatigue”

Panels C and D in Figure 3 are very similar, thus basically providing the same information. If these scales provide overlapping information, please consider removing one from the manuscript. For example, I would suggest removing the 20-point visual-analogue scale for determining "physical fatigue", because it was initially used to measure mental fatigue (lines 147 and 148).

Response:

We agree that there is an overlap of information. To prevent overlap, reuslts from the 20-point visual-analogue scale (panel D) have been removed from the manuscript. Amendments have been made to the revised manuscript wherever there is mention of fatigue, to clarify that statements are made based on the SURG-TLX results and not from the 20-point visual-analogue scale. 

From the materials and methods section, the following sentences have been removed: “Physical fatigue was measured using a modified, 0-20 point visual-analogue scale, initially used to measure mental fatigue. The scale ranges from ‘no fatigue, full energy levels’ to ‘extremely fatigued.’”

Within the results section, scores, main effects and interactions for fatigue (from the 20-point visual analogue scale, have been removed. 

The discussion on responses from the 20-point visual analogue scale for fatigue has now been removed from the section that addresses individual perceptual responses and has been moved to the section of the discussion that addresses the SURG-TLX questionnaire, under the heading/domain of ‘physical demand’ Lines 365-370 now read: “Notably, the physical demand of the surgical task, i.e., how physically fatiguing the procedure was, was higher in the HOT compared to CON theatre. Increased levels of fatigue reported during surgery could be of significant consequence in the health care industry, since fatigue is associated with an increased risk of medical errors [46, 47], carelessness among health care workers [48], and impairment to physical and mental performance during simulated medical work [49]”.

Line 409 of the manuscript now reads: “Our results suggest that it would be beneficial to consider fatigue/the physical demand of tasks and mental workload in the work design for major burns (heated surgeries)”. 

While reading the Results section, I noticed that the authors could clearly describe the main effect of time or the main effect of ambient temperature. However, I missed the information about the time x ambient temperature interactions. This issue is evident when analyzing data presented in figures and Table 1. Therefore, please include the information about interactions in the revised manuscript.

Response:

All information about interactions has now been included in the revised manuscript. The following additions have been made;

On page 9, lines 213-214, a sentence regarding the interaction between time and theatre temperature on counting latency has been added and the section now reads: There was no interaction between theatre temperature and time on counting latency (p = .203; Fig 1).

On page 9, lines 216-217, a sentence regarding the interaction between time and theatre temperature on the number of correct responses for the counting task has been added and the section now reads: “There was no interaction between theatre temperature and time on the number of correct responses (p = .757; Table 1)”.

On page 10, lines 232-233, a sentence regarding the interaction between time and theatre temperature on recall latency has been added and the section now reads: “There was no interaction between theatre temperature and time on recall latency (p = .757; Fig 2)”.

On page 10, lines 235-236, a sentence regarding the interaction between time and theatre temperature on number of correct responses for the recall task has been added and the section now reads: “There was no interaction between theatre temperature and time on number of correct responses (p = .828; Table 1)”.

On page 10, lines 238-239, a sentence regarding the interaction between time and theatre temperature on overall counting span score has been added and the section now reads: “There was no interaction between theatre temperature and time on overall counting span score (p = .949; Table 1)”.

On page 11, lines 252-254, a sentence regarding the interaction between time and theatre temperature on manual dexterity has been added and the section now reads: “There was no interatcion between theatre temperature and time on manual dexterity in either the dominant hand (p = .428) or the non-dominant hand (p = .949)”. 

On page 11, lines 262-266, a sentence regarding the interaction between time and theatre temperature on perceptual responses has been added and the section now reads: “There was an interaction between theatre temperature and time on TS (p = .019), TC (p = .047), and RPE (p < .001), which indicated similar perceptual responses for TS and RPE in CON and HOT pre-surgery (p > .924), but a higher TC score pre-surgery in HOT (11 ± 3) than CON (8 ± 4; p = .025)”.

On page 12, lines 288-289 a sentence regarding the interaction between time and theatre temperature on core temperature and heart rate has been added and the section now reads: “There was no interaction between theatre temperature and time on TCORE (p = .138) or HR (p = .700)”.

On page 13, lines 302-303 a sentence regarding the interaction between time and theatre temperature on USG scores has been added and the section now reads: “There was no interaction between theatre temperature and time on USG scores (p = .138)”. 

Although the figures were well elaborated, they can still be improved in several ways. First, most font sizes should be increased, especially in Figures 3, 4, and 6. Second, the white space between panels should be reduced in all figures. For example, the letters indicating the panels can be placed closer to or even inside these panels. Third, PLoS One is a journal published online, and it does not charge an additional tax to publish color figures. The manuscript will benefit from using blue symbols for the control condition and red symbols for the hot condition.

Response:

We agree that the figures could be improved and so they have been amended accordingly. Font size in all figures has been increased. The letters indicating panels have now been placed inside the figures, thus reducing white space on all figures. The scatter plots have been modified to aligned dot plots for visual aesthetic purposes and colours have been added to all figures (blue symbols for the CON condition and red for the HOT condition). 

Figure 4. The authors may want to present a panel with the overall score for the task load index (or workload), thus reducing the white space in this figure.

Response:

Thank you for your suggestion. We agree that the figure could be improved and so the mean scores for each dimension of the task load index have been presented in a new figure, reducing the white space. The figure legend, on line 280, for this figure has been altered to read “Scores for each dimension of the task load index in CON (n=22) and HOT (n=18) surgeries; mental demand (Men), physical demand (Phy), temporal demand (Tem), task complexity (Task), situation stress (Sit), distractions (Dist), and frustration (Frust). **indicates significant difference between conditions (p < .01); ***indicates significant difference between conditions (p <.001). Results are presented as mean ± SEM”.

MINOR COMMENTS 

Abstract, line 38. When mentioned for the first time, it may be interesting to emphasize that authors are referring to the term "body core temperature". After that, writing only "core temperature" or using the corresponding abbreviation is okay. This suggestion is also valid for the regular text (line 126).

Response:

In the abstract, the first sentence from lines 35-36 now reads: “Raising the ambient temperature of the operating theatre is common practice during burn surgeries to maintain the patient’s core body temperature”.

Abstract, line 46. Please define the meaning of the “RH” abbreviation before its first appearance.

Response:

Thank you for this observation. In the abstract, upon first appearance of the abbreviation RH (line 43), the term “relative humidity” has been inserted. 

Introduction, line 82. Please consider adding “not simulated” in the following sentence to improve clarity: “measured in a real-world (not simulated) context”.

Response:

The sentence in the introduction on line 80 has now been altered to read: “measured in a real-world (not simulated) context”.

Methods. Comparisons in lines 104 to 108 should be moved to the Results section. If the authors decide to maintain this information in the methods (although not ideal), please explain the meaning of the data (i.e., mean ± SEM) and include the statistical test used to generate the p-values reported.

Response:

We agree and have moved the aforementioned statements regarding duration of surgery and patient TBSA to the results section and have clarified the meaning of the data by re-wording the sentence. The first paragraph of the results section from lines 201-204 now reads: “Environmental conditions were 24.0±1.1°C, 45±6% RH for the CON trials, and 30.8±1.6°C, 39±7% RH for the HOT trials. Surgery duration was not different between conditions (CON: 141 ± 50 min, HOT: 158 ± 51 min; p = .287). Burn injury TBSA of patients was not different between conditions (CON: 8±13%, HOT: 20±7%; p = .053).” The statistical test used to generate the results is in the statistical analysis write up, before the mention of p values regarding surgery duration and TBSA. 

Methods, line 107. The average total body surface area was less than 20% (i.e., 8 ± 13%) during surgery in control conditions. This means that surgery was not necessarily a major burn surgery. Please amend the sentence in line 384.

Response:

Thank you for your comment. For clarification, the major burns are those conducted in the heat and are typically ≥ 20% TBSA. This has been noted in the introduction. The first sentence of the introduction, page 3 line 59, has now been amended to read: “Major burn surgeries, usually ≥ 20% total body surface area). The burns conducted in the CON condition are not major burns, they are the minor elective surgeries performed on burns usually < 20% TBSA. The sentence on line 410 is referring to the HOT surgeries, which are major burns. The term ‘heated surgeries’ has been added for clarity on line 410 and the sentence from lines 409-410 now reads: “Our results suggest that it would be beneficial to consider fatigue/the physical demand of the task and mental workload in the work design for major burns (heated surgeries)”.

Results, lines 190 to 193. To improve the description, the authors should mention the menstrual phase of the following two women surgeons: “… one was only tested once, and one was in the same menstrual cycle phase during testing sessions”.

Response:

Upon re-examination of the data, we have identified that one participant may have been in a different menstrual cycle phase during her testing sessions and so this has been added in to the text. The female participant who was tested once only was identified to be in the follicular phase of the menstrual cycle. All information has been added to the written text and the results section from lines 204-208 and the section now reads: “Of the seven females tested, three were post-menopausal, two were using an intrauterine device which meant that their menstrual cycle phase was unidentifiable, one was only tested once in the follicular phase, and one was in the follicular phase during the first testing session and the luteal phase during the second”.

Table 1, lines 209 to 211. Please consider replacing “49 ± 9.0” with “49 ± 9”.

Response:

The text in the table (lines 226-227) has been adjusted accordingly to read “49 ± 9”.

Discussion, line 306. Please indicate the number of the reference instead of the publication year.

Response:

Thank you for this observation. On line 328, the reference has been added. The sentence now reads: “Improvements in manual dexterity in the dominant hand over time, as found in this study, were also seen by Palejwala and colleagues [16]...”

Discussion, line 376. Please consider citing the following reference – doi: 10.1080/23328940.2020.1868386 – or any other relevant reference.

Response:

Thank for providing the resource above. The reference has been incorporated into the discussion session, following the mention of cooling technologies for staff. Another relevant reference discussing head cooling has also been added to this section. The section on lines 399-404 now reads: “Alleviating symptoms of heat strain in burn surgery staff should be a priority and could be achieved by taking small breaks during surgery, using underbody warming devices for the patient (warming mattresses) as opposed to heating the OT, or cooling technologies for staff. For example, head cooling caps [54] and cooling vests [55] have been found to lower perceptual heat strain and may be able to do the same for surgery staff working in the heat, thereby lowering their mental workload”.

---

## [Decision Letter · Decision Letter 1]

19 May 2023

PONE-D-23-05820R1Higher operating theatre temperature during burn surgery increases physiological heat strain, subjective workload, and fatigue of surgical staffPLOS ONE

Dear Dr. Zehra Palejwala,

Thank you for submitting your manuscript to PLOS ONE. After careful consideration, we feel that it has merit but does not fully meet PLOS ONE’s publication criteria as it currently stands. Therefore, we invite you to submit a revised version of the manuscript that addresses the points raised during the review process.

The authors have addressed all the comments made by the reviewers satisfactorily. Indeed, the authors were highly responsive to these comments. Thank you! As a result, the revised manuscript was much improved compared to its first version. However, a few minor modifications should be made before the study is ready for publication; please see the Academic Editor’s comments at the end of this letter.

We look forward to receiving your revised manuscript.

Kind regards,

Samuel Penna Wanner, Ph.D.

Academic Editor

PLOS ONE

Journal Requirements:

Additional Editor Comments:

Minor points should be addressed before the manuscript is ready for publication. If the authors address these points (they will do), the manuscript will be accepted without an additional round of external reviews.

1) Abstract, line 50. Please consider referring to the HOT condition as approximately 31�C heat, not 30�C.

2) Methods, line 145. Please consider replacing “TCORE and HR measurement was taken as soon” with “TCORE and HR measurements were taken as soon”.

3) Please describe the results in a better way. For example, in the paragraph between lines 211 and 217, the authors could merge the two sentences about interaction into one sentence. Suggestion: “There was no interaction between theatre temperature and time on counting latency (p = .203; Fig 1) or time on the number of correct responses (p = .757; Table 1)”.

3.1) Similarly, in the paragraph between lines 230 and 239, the authors could merge the three sentences about interaction into only one sentence. Suggestion: “There was no interaction between theatre temperature and time on recall latency (p = .821; Fig 2), number of correct responses (p = .828; Table 1), or overall counting span score (p = .949; Table 1).

4) Results, lines 262 to 266. It is challenging to understand the information in the following sentence: “There was an interaction between theatre temperature and time on TS (p = .019), TC (p = .047), and RPE (p < .001), which indicated similar perceptual responses for TS and RPE in CON and HOT pre-surgery (p > .924), but a higher TC score pre-surgery in HOT (11 ± 3) than CON (8 ± 4; p = .025).” Please amend this sentence to improve clarity. The authors may want to focus on post-surgery rather than on pre-surgery data when describing the findings concerning these three perceptual variables. In the case of TC, it is also valid to comment that a difference already exists before surgery.

5) Please indicate where the Discussion section starts. The revised manuscript lacks the heading of the discussion section.

6) The figures are not shown in an appropriate order in the pdf file of the revised manuscript. For example, figure 6 comes before figures 1 and 2. Please amend this issue to avoid mistakes in the manuscript that will be published.

Reviewers' comments:

Reviewer's Responses to Questions

**Comments to the Author**

1. If the authors have adequately addressed your comments raised in a previous round of review and you feel that this manuscript is now acceptable for publication, you may indicate that here to bypass the “Comments to the Author” section, enter your conflict of interest statement in the “Confidential to Editor” section, and submit your "Accept" recommendation.

Reviewer #1: (No Response)

Reviewer #2: All comments have been addressed

Reviewer #3: All comments have been addressed

2. Is the manuscript technically sound, and do the data support the conclusions?

Reviewer #1: Yes

Reviewer #2: Yes

Reviewer #3: Yes

3. Has the statistical analysis been performed appropriately and rigorously? 

Reviewer #1: Yes

Reviewer #2: Yes

Reviewer #3: Yes

4. Have the authors made all data underlying the findings in their manuscript fully available?

Reviewer #1: Yes

Reviewer #2: (No Response)

Reviewer #3: Yes

5. Is the manuscript presented in an intelligible fashion and written in standard English?

Reviewer #1: Yes

Reviewer #2: Yes

Reviewer #3: Yes

6. Review Comments to the Author

Reviewer #1: I was very pleased with the authors' review of the article and their responses to my questions. Congratulations on the nice article!

Reviewer #2: (No Response)

Reviewer #3: The authors addressed my comments satisfactorily. Congratulations!

Thank you for the opportunity to review this manuscript.

Kind Regards.

7. PLOS authors have the option to publish the peer review history of their article (what does this mean?). If published, this will include your full peer review and any attached files.

Reviewer #1: No

Reviewer #2: **Yes: **Alexandre SR Hudson

Reviewer #3: No

---

## [Author Response · Author response to Decision Letter 1]

21 May 2023

GENERAL COMMENTS

The authors have addressed all the comments made by the reviewers satisfactorily. Indeed, the authors were highly responsive to these comments. Thank you! As a result, the revised manuscript was much improved compared to its first version. However, a few minor modifications should be made before the study is ready for publication; please see the Academic Editor’s comments at the end of this letter.

Reviewer #1: I was very pleased with the authors' review of the article and their responses to my questions. Congratulations on the nice article!

Reviewer #3: The authors addressed my comments satisfactorily. Congratulations! Thank you for the opportunity to review this manuscript. Kind Regards.

Response: We thank the reviewers for their positive feedback on the manuscript.

JOURNAL REQUIREMENTS:

Response: 

The reference list has been reviewed and is complete and correct. No retracted papers have been cited in the mnuscript. 

MINOR REVISIONS

Abstract, line 50. Please consider referring to the HOT condition as approximately 31�C heat, not 30�C.

Response: 

Lines 49-51 of the abstract now read: “Cognitive function and manual dexterity were mainained in hot conditions, suggesting that operating in approximately 31�C heat is a safe approach for patient treatment. 

Methods, line 145. Please consider replacing “TCORE and HR measurement was taken as soon” with “TCORE and HR measurements were taken as soon”.

Response: 

Line 145 of the methods section now reads: “TCORE and HR measurements were taken as soon as surgery commenced...” 

Please describe the results in a better way. For example, in the paragraph between lines 211 and 217, the authors could merge the two sentences about interaction into one sentence. Suggestion: “There was no interaction between theatre temperature and time on counting latency (p = .203; Fig 1) or time on the number of correct responses (p = .757; Table 1)”.

Response: 

Lines 215-217 of the results section now read: “There was no interaction between theatre temperature and time on counting latency (p = .203; Fig 1) or the number of correct responses (p = .757; Table 1)”. 

Similarly, in the paragraph between lines 230 and 239, the authors could merge the three sentences about interaction into only one sentence. Suggestion: “There was no interaction between theatre temperature and time on recall latency (p = .821; Fig 2), number of correct responses (p = .828; Table 1), or overall counting span score (p = .949; Table 1).

Response: 

Lines 235-237 of the results section now read: “There was no interaction between theatre temperature and time on recall latency (p = .821; Fig 2), number of correct responses (p = .828; Table 1), or overall counting span score (p = .949; Table 1).

Results, lines 262 to 266. It is challenging to understand the information in the following sentence: “There was an interaction between theatre temperature and time on TS (p = .019), TC (p = .047), and RPE (p < .001), which indicated similar perceptual responses for TS and RPE in CON and HOT pre-surgery (p > .924), but a higher TC score pre-surgery in HOT (11 ± 3) than CON (8 ± 4; p = .025).” Please amend this sentence to improve clarity. The authors may want to focus on post-surgery rather than on pre-surgery data when describing the findings concerning these three perceptual variables. In the case of TC, it is also valid to comment that a difference already exists before surgery.

Response: 

Thank you for your comment. To clarify, the post-surgery perceptual responses have been explained in detail in respect to the main effect for time. A sentence has been added to the discussion of the interaction to explain the post-surgery comparison between CON and HOT. The statement regarding the interaction discusses pre-surgery as that is where the significant post-hocs were found. To improve clarity, lines 260-264 of the manuscript now read: “There was an interaction between theatre temperature and time on TS (p = .019), TC (p = .047), and RPE (p < .001). The interaction supported that scores post-surgery were higher in HOT than CON for all perceptual measures (p < .001). It also revealed that ratings of TS and RPE were the same pre-surgery in CON and HOT (p > .924), but a difference in ratings of TC existed before surgery (HOT: 11 ± 3, CON: 8 ± 4; p = .025)”.

Please indicate where the Discussion section starts. The revised manuscript lacks the heading of the discussion section.

Response: 

The heading “Discussion” has now been added to line 314 of the manuscript, to indicate where the discussion section starts. 

The figures are not shown in an appropriate order in the pdf file of the revised manuscript. For example, figure 6 comes before figures 1 and 2. Please amend this issue to avoid mistakes in the manuscript that will be published.

Response: 

The figures have now been uploaded in the appropriate order in the PDF file of the revised manuscript.

---

## [Editor Report · Decision Letter 2]

23 May 2023

Higher operating theatre temperature during burn surgery increases physiological heat strain, subjective workload, and fatigue of surgical staff

PONE-D-23-05820R2

Dear Dr. Zehra Palejwala,

We’re pleased to inform you that your manuscript has been judged scientifically suitable for publication and will be formally accepted for publication once it meets all outstanding technical requirements.

Kind regards,

Samuel Penna Wanner, Ph.D.

Academic Editor

PLOS ONE

Additional Editor Comments (optional):

The authors have satisfactorily addressed all six comments made by this Academic Editor. As a result, the revised manuscript is ready for publication. Congratulations on the excellent study!
---

## [Editor Report · Acceptance letter]

25 May 2023

PONE-D-23-05820R2 

Higher operating theatre temperature during burn surgery increases physiological heat strain, subjective workload, and fatigue of surgical staff 

Dear Dr. Palejwala:

I'm pleased to inform you that your manuscript has been deemed suitable for publication in PLOS ONE. Congratulations! Your manuscript is now with our production department. 

Kind regards, 

on behalf of

Dr. Samuel Penna Wanner 

Academic Editor

PLOS ONE